# Decision-Making Method for Mine Cable Insulation Monitoring and Grounding Fault Diagnosis

Chen Feng [1], Pingfeng Ye [1,*], Yanying Sun [2], Jingrui Li [1], Xiangyu Zang [1] and Chenhao Sun [1]

1   College of Electrical Engineering and Automation, Shandong University of Science and Technology, Qingdao 266590, China
2   School of Mechanical Electronic and Information Engineering, China University of Mining and Technology (Beijing), Beijing 100083, China
*   Correspondence: ypf@sdust.edu.cn

**Abstract:** Real-time monitoring of the power cable state has tremendous significance for ensuring the safe and economic operation of mine power distribution systems. However, due to the harsh conditions of underground coal mines, it is difficult for the cable monitoring system operating in underground coal mines to carry out large-scale calculations to diagnose the grounding fault of the cable. Additionally, there are many types of cable grounding faults, such as single grounding fault and two-phase ground fault. Therefore, how to determine the type of grounding fault quickly through effective calculations and alarms to select the grounding cable is always a difficult task. In this study, to reduce the complexity of cable insulation state classification, we develop a novel classification method based on the decision tree algorithm. Concerning the zero-sequence network under different insulation conditions, the first calculation's positive and negative values were generated to identify whether the cable insulation was symmetrical. Then, the insulation degradation phase is identified by the relationship between the three-phase voltage phase angle and the current difference variation between the beginning and end of the line. By substituting the correlation quantities collected by a wide-area synchronous measurement system into different equations, the whole grid's decision tree was constructed in different insulation states. Then, the insulation state of each line was evaluated according to the conductance value. The effectiveness of the proposed method was verified using a 35/6 kV mine power distribution system model based on the MATLAB/Simulink platform. The test results illustrate that the method can accurately diagnose the power cable insulation state based on the decision tree of whether the grid three-phase loads operate in an ungrounded mode or not.

**Keywords:** mine cable insulation degradation; online insulation monitoring; decision tree method



## 1. Introduction

Power cables are widely used in coal mine power distribution systems. However, due to the narrow space of the coal mine, the mine cables are mostly laid along the roadways, and they are vulnerable to external forces such as squeezing and collision, which damages the insulation of the cables and causes a cable grounding fault. When the cable insulation deteriorates, it may cause an electric shock or cable firing. This situation is very dangerous in coal mines and may cause more serious coal mine accidents. Therefore, it is significant to perform insulation monitoring on the mine cables in real-time to detect and eliminate faults in time [1–6].

The existing online cable insulation monitoring methods include the partial discharge, DC superposition, ground line current, and dielectric loss methods [1,6–17]. The partial discharge signal is difficult to measure and is not suitable for harsh conditions [9,10]. The DC superposition method requires an additional source and cannot measure the distributed capacitance of the cable to the ground [11,12]. The grounding current and dielectric loss methods are inaccurate because they use empirical values to evaluate the cable insulation

state that proposes that two branches be selected for grounding experiments. The insulation parameters of all branches can be calculated by collecting the zero-sequence current of each branch and the zero-sequence voltage of the bus. However, this method does not consider the insulation asymmetry which may exist in the cable itself [13–17]. In order to make up for the shortcomings of previous methods, many scholars have proposed some new methods for cable insulation diagnosis in recent years. For example, in the literature [18], low-frequency signals are coupled into cables to monitor the insulation of cables, the leakage current and voltage data of cables are extracted, and the insulation impedance is calculated by the fast Fourier algorithm. Ref. [19] proposed a monitoring scheme method based on leakage current measurements at a selected monitoring frequency, and an aging feature extraction method based on principal component analysis (PCA), which can estimate the severity of insulation aging. With the development of computer technology, more and more scholars use data-driven machine learning algorithms in cable fault diagnosis [20–31]. Ref. [20] proposed a cable fault diagnosis knowledge base based on machine learning. By collecting the wavelet entropy of cable fault signals, the noise change relationship of the fault signal is reflected, and the noise energy threshold of the cable fault signal is determined. The fuzzy decision matrix of the cable fault diagnosis knowledge base is established by machine learning. Ref. [21] established fault criteria and a database by simultaneously measuring two current maps in coaxial cables, and detected cable faults by analyzing changes in characteristic parameters of the maps. Ref. [22] introduced a fault severity prediction method based on a one-dimensional convolution residual network without prior cable knowledge. By correlating the channel transfer function (CTF) with the fault information, a pilot-based deep learning network that can extract fault features more accurately emerged.

However, the above methods are aimed at the fault diagnosis of cables in the common power grid. Compared with the ordinary power grid, underground coal mine power grids have many particularities. The fault diagnosis method of coal mine cables cannot follow the fault diagnosis method of common cables. Due to the harsh working environment of downhole equipment, large-scale data communication cannot be realized. Underground mine cable monitoring systems, generally perform in situ monitoring and directly calculate diagnoses and alarms. Generally, there are only small current grounding monitoring systems in cable monitoring systems. The data types and fault samples obtained are very few. In this environment, DSP is generally used as the computing core of cable fault diagnosis equipment [23–25]. Therefore, it is difficult to use machine learning algorithms for fault diagnoses of underground coal mine cables. Additionally, for the coal mine power grid, the most urgent need is to quickly diagnose the grounding fault of the cable and quickly cut off the fault, so as to prevent accidents. How to judge the type of grounding fault quickly through effective calculations under such harsh conditions, and to select the grounding cable is a difficult point at present. In addition, the above method of judging the cable insulation state is by one single calculation model. For different types of insulation deterioration, the relationship between the zero-sequence component of cable and insulation parameters is different, that is, the calculation model of cable insulation state is different. Only one model as the basis for the diagnosis of cable insulation status cannot accurately reflect the real insulation level of the cable, and there is the possibility of misjudgments.

In view of the problems above, a new method for cable grounding fault monitoring is proposed. First, by modeling the zero-sequence network of cables with single-phase and two-phase insulation degradation, a method applicable for pattern recognition analysis is established. Based on this, a diagnostic decision tree based on the zero-sequence components and insulation parameters to diagnose the cable insulation state is proposed, which can judge multiple insulation faults under complex conditions. Finally, a simulation model is built on the MATLAB/Simulink platform to verify the effectiveness of the proposed method. The simulation results illustrate that our method can accurately diagnose the power cable insulation state.

The rest of this paper is organized as follows. In Section 2, according to the different types of cable insulation deterioration, the criterion model of insulation state diagnosis is established, and the principle of cable insulation diagnosis is described. In Section 3, the construction method of the decision tree is introduced. In Section 4, the effectiveness of the proposed method is verified using a simulation.

## 2. Materials and Methods

The equivalent operation circuit of the mine power grid is shown in Figure 1a. If the zero-sequence voltage $U_0$ appears due to parameter asymmetry, the voltage at the system's neutral point is equal to the zero-sequence voltage. The transformer's neutral point adopts the ineffective grounding method (not grounded or grounded via the arc suppression coil), and the neutral point of the three-phase load for each branch is grounded.

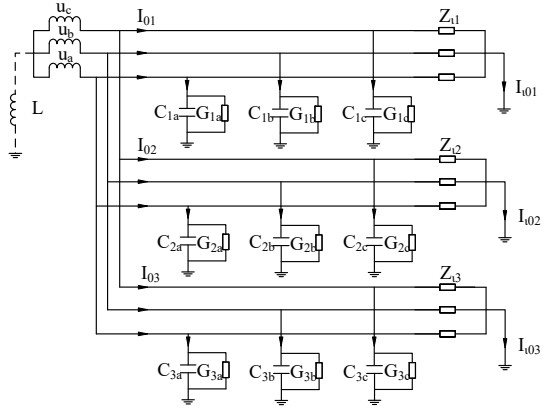

(a) The equivalent operation circuit of the power grid

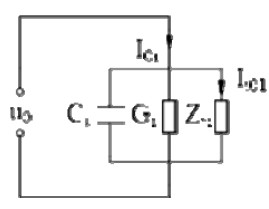

(b) The zero-sequence circuit of the symmetrical branch

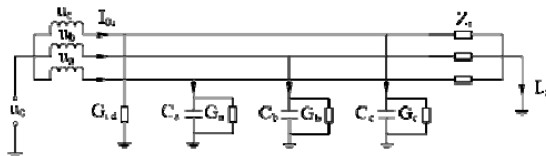

(c) The equivalent operation circuit of the faulty branch with single-phase insulation degradation

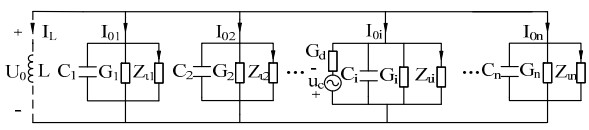

(d) The zero-sequence circuit of the system with single-phase insulation degradation

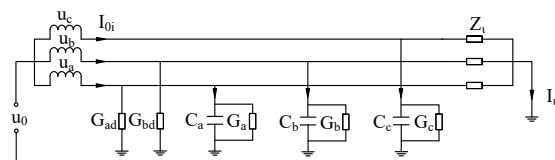

(e) The equivalent operation circuit of a branch with two-phase insulation degradation

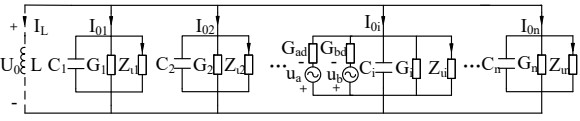

(f) The zero-sequence circuit of a system with two-phase insulation degradation

**Figure 1.** The different circuits.

In Figure 1a, $Z_{i1}$, $Z_{i2}$, $Z_{i3}$, $(i = 1, 2 \ldots .n)$ are the arc suppression coil reactances; $G_{iA}$, $G_{iB}$, $G_{iC}$, $(i = 1, 2 \ldots .n)$ are the three phase insulation conductances for each branch; $C_{iA}$, $C_{iB}$, $C_{iC}$, $(i = 1, 2 \ldots .n)$ are the three phase distributed capacitances for each branch; $I_{0i}$ $(i = 1, 2 \ldots .n)$ is the zero-sequence current of each branch; $Z_{li}$ $(i = 1, 2 \ldots .n)$ is the impedance of load for each branch; and $I_{l0i}$ $(i = 1, 2 \ldots .n)$ is the zero-sequence current flowing through the load into the earth. Figure 1b shows the zero-sequence circuit of the branch with symmetrical insulation parameters.

According to Kirchhoff's Current Law, the zero-sequence voltage and current of a three-phase symmetrical line satisfy

$$\dot{I}_{0i} - \dot{I}_{l0i} = Y_i \dot{U}_0 = G_i \dot{U}_0 + j\omega C_i \dot{U}_0 \tag{1}$$

where, $Y_i = Y_a + Y_b + Y_c$, $Y_a = G_a + jwC_a$, $Y_b = G_b + jwC_b$, and $Y_c = G_c + jwC_c$, $Y_a = Y_b = Y_c$.

In the case of single-phase and two-phase deterioration of cable insulation, the equivalent circuit model of the degraded insulation line is different because of the difference of the zero-sequence voltage value in the zero-sequence network. In order to comprehensively analyze the cable insulation deterioration, the characteristics of the two types of insulation degradation will be analyzed one by one.

### 2.1. Zero-Sequence Network of a Power System under Single-Phase Insulation Degradation

When single-phase insulation degradation occurs in the system, it is equivalent to adding a conductance $G_d$ between the faulty branch and the ground. Assuming that the faulty phase is phase C, its equivalent branch operation circuit is shown in Figure 1c. It can be seen from Figure 1c that zero sequence current appears in phase C. Additionally, the zero-sequence current of the faulty branch satisfies Equation (2).

$$\dot{I}_{0i} - \dot{I}_{l0i} = (\dot{U}_0 + \dot{U}_a)Y_a + (\dot{U}_b + \dot{U}_0)Y_b + (\dot{U}_c + \dot{U}_0)(Y_c + G_d) \tag{2}$$

Since the three-phase voltage meets equation $\dot{U}_a + \dot{U}_b + \dot{U}_c = 0$. Then, Equation (2) can be simplified as

$$\dot{I}_{0i} - \dot{I}_{l0i} = \dot{U}_0 Y_i + (\dot{U}_c + \dot{U}_0)G_d \tag{3}$$

Therefore, the zero-sequence circuit of a system with single-phase insulation degradation is shown in Figure 1d.

When the neutral point is not grounded, the relationship between the zero-sequence and phase voltages is

$$
\begin{aligned}
\dot{U}_0 &= -\frac{\dot{U}_c G_d}{G_\Sigma + j\left(\omega C_\Sigma - \frac{1}{\omega L}\right) + G_d} \\
&= \frac{\dot{U}_c G_d}{\sqrt{\left(G_\Sigma + G_d\right)^2 + \left(\omega C_\Sigma\right)^2}} e^{-j(\pi + \theta_1)}
\end{aligned}
\tag{4}
$$

$$\theta_1 = tan^{-1}\left(\frac{\omega C_\Sigma}{G_\Sigma G_d}\right) \tag{5}$$

When the neutral point is grounded via the arc suppression coil, the relationship between the zero-sequence and phase voltages is

$$
\begin{aligned}
\dot{U}_0 &= -\frac{\dot{U}_c G_d}{G_\Sigma + j\left(\omega C_\Sigma - \frac{1}{\omega L}\right) + G_d} \\
&= \frac{\dot{U}_c G_d}{\sqrt{\left(G_\Sigma + G_d\right)^2 + \left(\omega C_\Sigma\right)^2}} e^{-j(\pi + \theta_2)}
\end{aligned}
\tag{6}
$$

$$\theta_2 = tan^{-1}\left(\frac{\omega C_\Sigma - \frac{1}{\omega L}}{G_\Sigma + G_d}\right) \tag{7}$$

It can be seen from the above analysis that the grounding mode affects the zero-sequence voltage of the system when the single-phase insulation drops, but it does not affect the relationship between the zero-sequence component and the insulation parameters.

*2.2. Zero-Sequence Network of a Power System under Two-Phase Insulation Degradation—Abbreviations and Acronyms*

If two-phase insulation degradation occurs in the power system, the three-phase insulation conductance of the cable is not equal. Additionally, the zero-sequence network of the system is constant, and the asymmetry caused by the decrease of insulation causes the system to generate zero-sequence voltage. The equivalent operation circuit of the faulty branch is shown in Figure 1e.

According to Kirchhoff's current law, the zero-sequence current of the faulty branch satisfies Equation (8).

$$
\begin{aligned}
\dot{I}_{0i} - \dot{I}_{l0i} &= Y_a\left(\dot{U}_0 + \dot{U}_a\right) + (G_{db} + Y_b)\left(\dot{U}_0 + \dot{U}_b\right) + (G_{dc} + Y_c)\left(\dot{U}_0 + \dot{U}_c\right) \\
&= Y_i\dot{U}_0 + G_{db}\left(\dot{U}_0 + \dot{U}_b\right) + G_{dc}\left(\dot{U}_0 + \dot{U}_c\right)
\end{aligned}
\tag{8}
$$

The zero-sequence circuit of the whole system is shown in Figure 1f, the faulty branch is branch $i$, and the insulation of other branches is normal.

When the neutral point is not grounded, the relationship between the zero-sequence and the phase voltages is

$$
\begin{aligned}
\dot{U}_0 &= \frac{\dot{U}_b G_{db} + \dot{U}_c G_{dc}}{G_\Sigma + j\omega C_\Sigma + G_{bd} + G_{dc}} \\
&= \frac{\dot{U}_b G_{db} + \dot{U}_c G_{dc}}{\sqrt{\left(G_\Sigma + G_{db} + G_{dc}\right)^2 + \left(\omega G_\Sigma\right)^2}} e^{-j(\pi + \theta_1)}
\end{aligned}
\tag{9}
$$

$$
\theta_1 = tan^{-1}\left(\frac{\omega C_\Sigma}{G_\Sigma + G_{db} + G_{dc}}\right)
\tag{10}
$$

When the neutral point is grounded via the arc suppression coil, the relationship between the zero-sequence and phase voltages is

$$
\begin{aligned}
\dot{U}_0 &= -\frac{\dot{U}_a G_d}{G_\Sigma + j\left(\omega C_\Sigma - \frac{1}{\omega L}\right) + G_{db} + G_{dc}} \\
&= \frac{\dot{U}_a G_d}{\sqrt{\left(G_\Sigma + G_{db} + G_{dc}\right)^2 + \left(\omega C_\Sigma\right)^2}} e^{-j(\pi + \theta_2)}
\end{aligned}
\tag{11}
$$

$$
\theta_2 = tan^{-1}\left(\frac{\omega C_\Sigma - \frac{1}{\omega L}}{G_\Sigma + G_{db} + G_{dc}}\right)
\tag{12}
$$

It can be seen from the above analysis that the two grounding methods do not affect the relationship between the zero-sequence component of the branch and the insulation parameters. Additionally, the grounding mode affects the zero-sequence voltage of the system when the two-phase insulation drops, but it does not affect the relationship between the zero-sequence component and insulation parameters. In addition, there is three-phase cable insulation degradation. The three-phase insulation parameters of cable insulation are equal when the three-phase insulation degrades symmetrically. However, the probability of three-phase insulation degrading symmetrically in the power supply system is very small. Therefore, it is not discussed in this article.

According to the relationship between insulation parameters and the zero-sequence component, cable insulation degradation can be divided into symmetric insulation degradation and asymmetric insulation degradation. Additionally, the insulation asymmetry state can be further divided into single-phase and two-phase insulation degradation. It can be seen from Equation (4) to Equation (12) that the voltage and phase angle corresponding to different insulation faults are different. Therefore, the diagnosis of different types of cable insulation faults can be realized by monitoring the voltage and phase angle.

For the coal mine cable monitoring system, the real-time monitoring data are analyzed by a separate model for each type of fault, which not only increases the amount of calculations, but also affects the time of the system to judge the fault [23,24,31]. Therefore, we

propose a classification method from top to bottom using a decision tree to solve the cable grounding fault classification problem quickly and effectively. There are different formulas, which fit different modes, to calculate insulation parameters accurately. The fault is issued when the computed value of the insulation parameter is less than the setting value. In this way, the system only needs to carry out a group of logical judgment processes through the decision tree model, and can quickly find the fault cable. Therefore, the diagnosis system identifies the fault type and output insulation parameters online.

### 2.3. The Criterion for Insulation State Diagnosis

The total insulation of the cable is the parallel resistance of the three-phase insulation resistance. The cable insulation state can be roughly divided into three categories, namely, good, moderate attention, and close attention, whether in operation or not. According to the 6 kV cable insulation evaluation criteria, the classification table is shown in Table 1.

**Table 1.** The criterion of cable insulation evaluation.

| Object | Resistance | Evaluation | Classification | Operation |
|--------|-----------|------------|----------------|-----------|
| The total insulation | >10 | good | S1 | use |
| | 1~10 | moderate attention | S2 | use and attention |
| | <1 | high attention | S3 | replace |

### 2.4. Cable Insulation Diagnosis Monitoring System

The diagnosis and monitoring system is shown in Figure 2. It consists of four parts: a GPS unit, phasor measurement unit, communication unit, and control center [22–26]. The GPS unit provides a synchronous timing signal for the entire network, ensuring the data are synchronized with no space limit. GPS equipment in the mine transmits signals to the surface through optical fibers or an underground communication network, and then synchronizes the time through the surface GPS system and satellite system. This monitoring method has been proposed in Refs. [6,23,24]. The phasor measurement unit is installed in the field dispersedly, and the zero-sequence voltage of the bus, the zero-sequence current, the phase to ground voltage, and other useful data are synchronously collected. The measuring units of each branch are shown in Figure 2. During the phasor measurement process, the data sampling pulse is controlled by the GPS synchronous timing pulse signal [24–28]. The communication network transmits data to the control center for online analysis, then the control center calculates insulation parameters and generates control strategies for protection online.

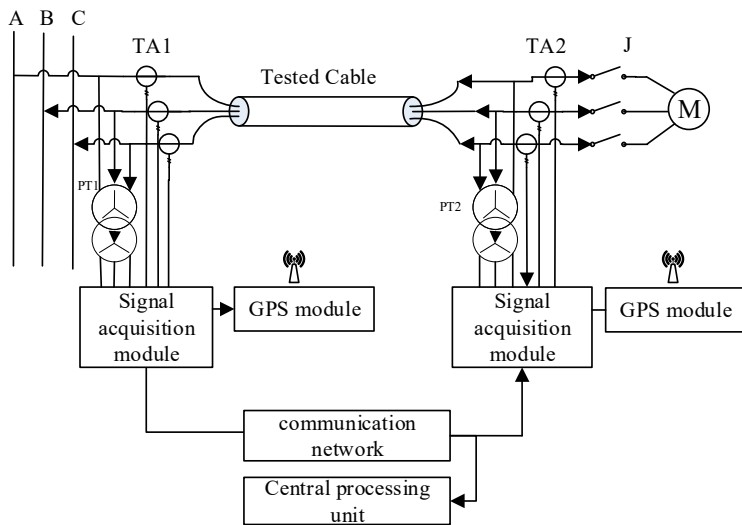

**Figure 2.** Diagram of phasor synchronization measurements.

### 2.5. Cable Insulation Diagnosis Method Base on Decision Tree

Cable insulation status can be divided into four types: insulation is normal, single-phase insulation degradation, two-phase insulation degradation, symmetrical three-phase insulation degradation. Cable grounding faults caused by the four types of cable insulation state are also divided into four types: normal state, single-phase grounding fault, two-phase grounding fault, and already three-phase grounding fault [1]. Through the previous analysis, it can be seen that the voltage and phase angle relationships of the four states are different, and it is an effective method to use classification algorithms to diagnose faults for this difference. The states of these four kinds of insulation are completely different, and the classification is complicated. In order to solve the complex classification problem of cable insulation quickly and effectively, a decision tree method is proposed to realize hierarchical classification from top to bottom.

Assuming all the lines' insulation parameters are symmetrical, the differential equation can be obtained, by

$$i_{0i} - i_{0li} = C_i \frac{du_0}{dt} + G_i u_0 \tag{13}$$

The zero-sequence voltage of the bus, the zero-sequence current at the beginning, and the zero-sequence current at the end of each branch are synchronously acquired. The derivative of the zero-sequence voltage can be calculated by

$$\frac{du_0}{dt} m = \frac{u_0(m+1) - u_0(m-1)}{2\Delta t} \tag{14}$$

where the $n$ is set as the sample data number and $\Delta t$ is set as the sample interval. Setting the equations by selecting sampled data at equal intervals:

$$\begin{pmatrix} \frac{du_0}{dt}(1) & u_0(1) \\ \frac{du_0}{dt}(2) & u_0(2) \\ \vdots & \vdots \\ \frac{du_0}{dt}(m) & u_0(m) \end{pmatrix} \cdot \begin{pmatrix} C_1 & C_2 & \cdots & C_n \\ G_1 & G_2 & \cdots & G_n \end{pmatrix} = \begin{pmatrix} i_{01}(1)' & i_{02}(1)' & \cdots & i_{0n}(1)' \\ i_{01}(2)' & i_{02}(2)' & \cdots & i_{0n}(2)' \\ \vdots & \vdots & \ddots & \vdots \\ i_{01}(m)' & i_{02}(m)' & \cdots & i_{0n}(m)' \end{pmatrix} \tag{15}$$

The $m$ is the number of the selected data in the equation. If one of the insulation parameters calculated by Equation (15) has a negative value, the insulation parameter of this line is asymmetrical. Conversely, it can be derived from Equations (1) and (2) that the change in the difference of the zero-sequence current $\Delta \dot{I}_i$ before and after single-phase insulation degradation is

$$\Delta \dot{I}_i = G_d \left( \dot{U}_c + \dot{U}_0 \right) + Y_i \left( \dot{U}_0^{'} - \dot{U}_0 \right) \tag{16}$$

The change in zero-sequence voltage is insignificant before and after the insulation degradation, so the zero-sequence current change can be ignored, and formula (17) turns into

$$\Delta \dot{I}_i \approx G_d \left( \dot{U}_c + \dot{U}_0 \right) = G_d \dot{U}_{cd} \tag{17}$$

$\dot{U}_{cd}$ is the relative voltage to the ground of phase C. It can be derived from Equation (18) such that the phase angle of the $\Delta \dot{I}_i$ of each branch is the same as the phase angle of the faulty phase's relative voltage to the ground. Therefore, this relationship can be used as a criterion to identify single-phase insulation degradation. When the zero-sequence current is at the beginning of the line, it can be judged that single-phase insulation degradation occurs on a particular line when the phase angle of the current difference variation is the same as one of the three relative phase voltages to the ground.

Simultaneously, a faulty branch with single-phase insulation degradation satisfies Equation (3), so the differential equation can be derived.

$$i_{0i} - i_{0li} = C_i \frac{du_0}{dt} + G_i u_0 + G_d u_{cd} \tag{18}$$

Substitute *m* sets of data collected by the wide-area synchronous measurement system into the equations. Additionally, the branch insulation degradation degree could be evaluated by comparing the calculated insulation parameters and the setting value. The decision subtree is shown in Figure 3b.

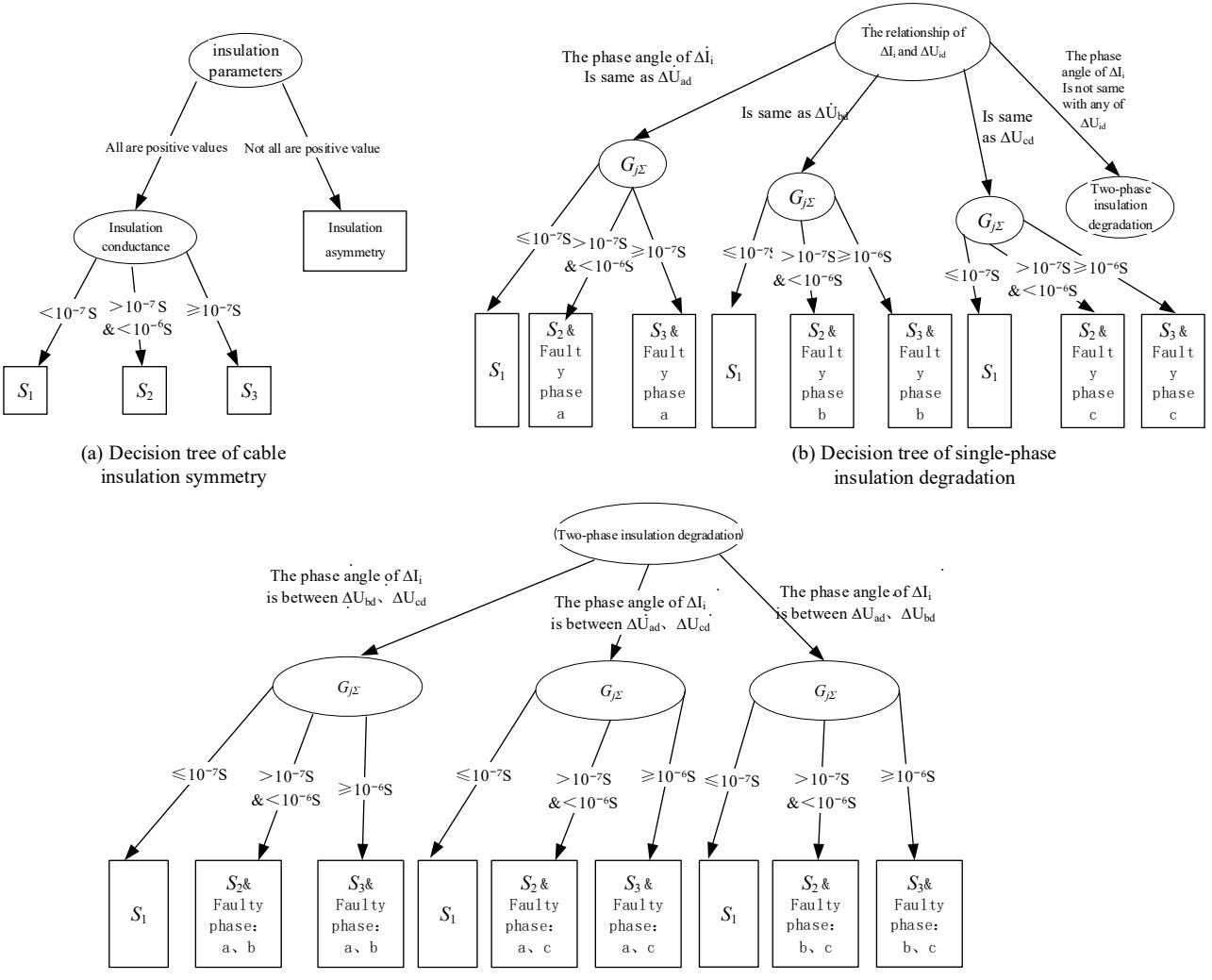

**Figure 3.** Decision tree.

When the two-phase insulation deteriorates (set to b and c phases), it can be derived from Equations (2) and (4) such that the variation of the zero-sequence current at the beginning and end of the branch is

$$\begin{aligned} \Delta \dot{I}_i &= G_{db}\left(\dot{U}_b + \dot{U}_0\right) + G_{dc}\left(\dot{U}_c + \dot{U}_0\right) + Y_i\left(\dot{U}_0' - \dot{U}_0\right) \\ &\approx G_{db}\dot{U}_{bd} + G_{dc}\dot{U}_{cd} \end{aligned} \tag{19}$$

Therefore, this characteristic can be used as a criterion when the phase angle of $\Delta \dot{I}_i$ of a certain branch is between the phase angle of the two phases' relative voltage to the ground. The monitoring will judge that this branch has two-phase insulation degradation and determine the faulty phase.

The degree of insulation degradation can be determined based on the calculated insulation parameters according to the equation. The decision subtree is shown in Figure 4c. The complete decision tree for the whole grid is shown in Figure 3c.

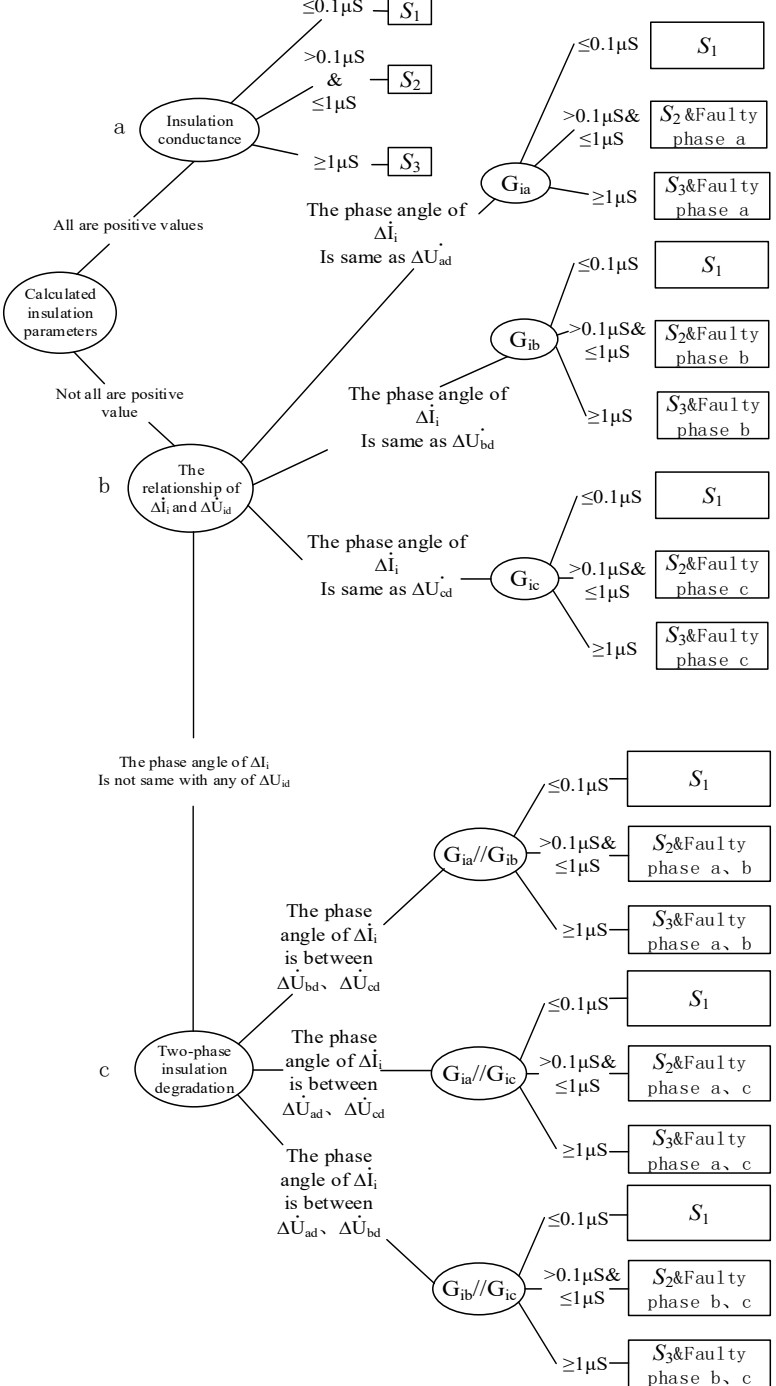

**Figure 4.** Integral decision tree: (**a**) Decision tree of cable insulation symmetry; (**b**) Decision tree of single-phase insulation degradation; (**c**) Decision tree of single-phase insulation degradation.

It should be noted that if the three-phase loads of the grid are operated in an ungrounded mode, the equation $i_0{}' = i_{0i} - i_{0l}$ should be substituted by $i_0 = i_{0i}$, because there will not be a current path for the zero-sequence current at the end of the line. Additionally, only the zero-sequence current at the head of each branch should be collected. Suppose a branch is connected to a substation of a lower voltage grade. In that case, the $i_0'$ should be the amount of current difference between branch zero-sequence current and zero-sequence current for all lines belonging to the substation.

The primary condition for subtree synthesis is the existence of the same properties. In the above three subtrees, single-phase insulation degradation and two-phase insulation degradation are both types of insulation asymmetry. The method to judge the deterioration of cable insulation is the phase relationship between the zero-sequence current conversion and the relative ground voltage. Therefore, three subtrees can be combined into one decision tree. The complete decision tree is shown in Figure 4.

## 3. Simulation Verification and Results

### 3.1. Construction of the Simulation Model

To verify the effectiveness of the insulation monitoring algorithm based on the decision tree, the 35/10 kV single radiating power supply system shown in Figure 5 was built on the MATLAB/Simulink platform. This is a very typical power supply network model in coal mines [1,6,24]. Table 2 shows the setting parameters of the insulation and load power of each branch. It can be seen from Section 2.3 that the two grounding methods do not affect the relationship between the zero-sequence component of the branch and the insulation parameters, so some types of the neutral points are grounded by the arc suppression coil (the overshoot compensation coefficient of the arc suppression coil is 1.5), and some types of neutral points are not grounded. Since the load imbalance does not affect the method, the model sets three-phase balanced loads for four branches. According to the cable insulation evaluation standard of 10 kV, when the calculated branch insulation conductance value is greater than 0.1 μS, the branch is judged to have slightly deteriorated insulation, and when the calculated branch conductance value is equal to or greater than 1 μS, the branch is judged to have seriously deteriorated insulation.

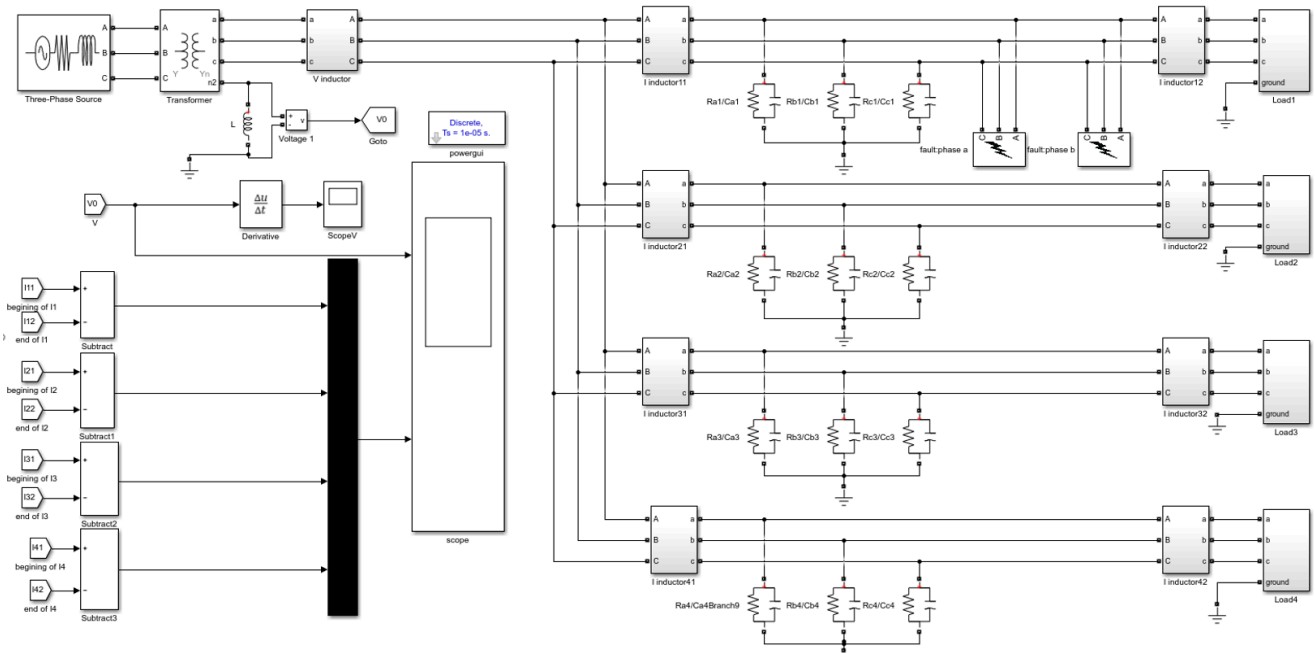

**Figure 5.** Simulation model.

**Table 2.** The setting parameters of insulation and load power for each branch.

| Branch Number | 1 | 2 | 3 | 4 |
|---|---|---|---|---|
| $C_i$ (F) | $0.059 \times 3$ | $0.117 \times 3$ | $0.422 \times 3$ | $0.750 \times 3$ |
| $G_i$ (μS) | $0.0025 \times 3$ | $0.005 \times 3$ | $0.018 \times 3$ | $0.032 \times 3$ |
| Load power (kW) | $100 \times 3$ | $100 \times 3$ | $100 \times 3$ | $100 \times 3$ |
| $L$ (H) | | 3.757 | | |

The model is taken from the power supply circuit diagram of an industrial and mining enterprise.

### 3.2. Analysis of Neutral Grounding System Simulation Results

The simulation time is 0.5 s, and four branches were selected as fault branches in turn. At 0.1 s, single-phase insulation degradation occurs on the faulty branch, and the phase a insulation conductance decreases to 0.5 μS. At 0.3 s, two-phase insulation degradation occurs on the faulty branch, and the phase b insulation of the faulty branch is reduced to 2 μS while the phase an insulation is maintained at 0.5 μS. Other branches have normal insulation during the simulation. While branch 1 is the faulty branch and the neutral point are grounded by the arc suppression coil, the waveform diagrams of the system zero-sequence voltage and the zero-sequence current difference between the beginning and the end of each branch are shown in Figure 6.

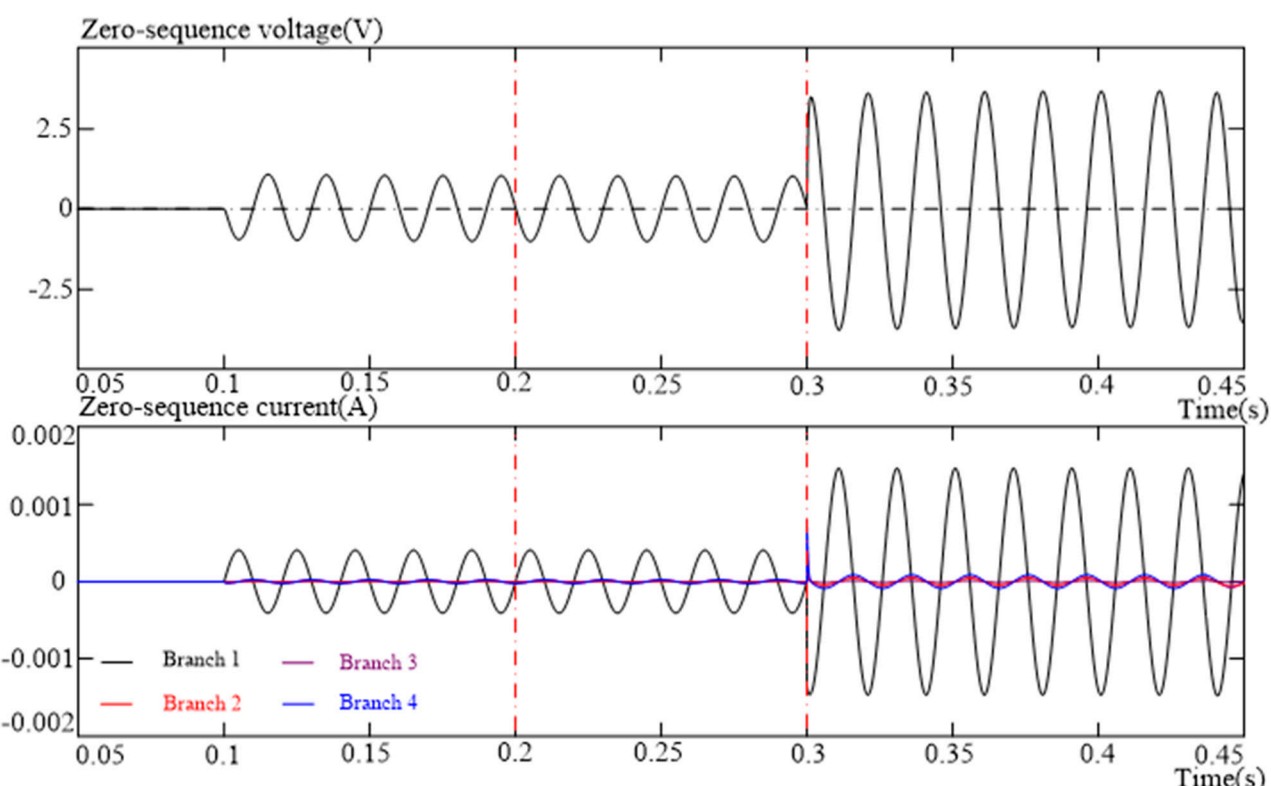

**Figure 6.** Zero-sequence voltage and current waveforms.

The insulation parameters of each branch calculated by Equation (18) are shown in Table 3. Among them, the insulation parameter of branch 1 has a negative value, and the insulation parameters of the other lines are all positive values, so it can be judged that the insulation asymmetry fault occurred on branch 1.

**Table 3.** Calculated insulation parameters for each line.

| Branch Number | 1 | 2 | 3 | 4 |
|---|---|---|---|---|
| Calculated $C_i$ (F) | $0.059 \times 3$ | $0.117 \times 3$ | $0.422 \times 3$ | $0.750 \times 3$ |
| Calculated $G_i$ (μS) | $0.0025 \times 3$ | $0.005 \times 3$ | $0.018 \times 3$ | $0.032 \times 3$ |
| Load power (kW) | $100 \times 3$ | $100 \times 3$ | $100 \times 3$ | $100 \times 3$ |

The system does not have zero-sequence voltage and current until 0.1 s, and because the system capacitance is continuously charged and discharged, the zero-sequence voltage has a DC component. At 0.3 s, the zero-sequence current of branch 1 has an obvious variation. The characteristics of the waveform diagrams prove that the simulation model is correct.

The calculated phase angle of $\triangle \dot{I}_i$ and the $\dot{U}_{ad}$, $\dot{U}_{bd}$, $\dot{U}_{cd}$ of branch 1 around the time when the single-phase insulation degradation occurred are shown in Table 4.

**Table 4.** Calculated phase angle of $\triangle \dot{I}_i$ and the $\dot{U}_{ad}$, $\dot{U}_{bd}$, $\dot{U}_{cd}$ of branch 1.

| Phasor | $\Delta \dot{I}_i$ | $\dot{U}_{ad}$ | $\dot{U}_{bd}$ | $\dot{U}_{cd}$ |
|---|---|---|---|---|
| Phase angle | −0.3 | 0 | 240 | 120 |
| The judgment of faulty phase | | a | | |

The calculations show that the phase angle of the current difference rate between the beginning and end of branch 1 is the same as the phase angle of phase c relative voltage to ground. Therefore, it can be determined that the faulty phase is phase c of branch 1. Substituting the sampled value of branch 1 into Equation (17), the results and calculation error are shown in Table 5.

**Table 5.** Calculated insulation parameters (numerical value and calculation error) of branch 1.

| Insulation Parameters | $C_1$ (μF) | $G_1$ (μS) | $G_d$ (μS) |
|---|---|---|---|
| Numerical value | 0.180 | 0.0078 | 0.500 |
| Calculation error | 1.6% | 4% | 0 |

It can be judged that the insulation of the branch 1 phase a is lower, and the calculation error is smaller. At 0.3 s, the insulation parameters of each line calculated by Equation (13) are shown in Table 6.

**Table 6.** Calculated insulation parameters of branch 1 at 0.3 s.

| Branch Number | 1 | 2 | 3 | 4 |
|---|---|---|---|---|
| Calculated $C_i$ (F) | −0.813 | 0.361 | 1.298 | 2.308 |
| Calculated $G_i$ (μS) | −0.328 | 0.015 | 0.054 | 0.096 |
| The judgment of insulation state | asymmetrical | symmetrical and good | symmetrical and good | symmetrical and good |

The insulation parameters of branch 1 have negative values, and the insulation parameters of branch 1 to branch 4 have positive values. It can be judged that insulation asymmetry occurred on branch 1. Therefore, it can be judged that branch 1 has asymmetrical insulation. The phase angle of $\Delta \dot{I}_1$ and the relative voltage to ground of the three phases are shown in Table 7.

**Table 7.** Calculated phase angle of $\Delta \dot{I}_3$ and the $\dot{U}_{1ad}$, $\dot{U}_{1bd}$, and $\dot{U}_{1cd}$.

| Phasor | $\Delta \dot{I}_1$ | $\dot{U}_{1ad}$ | $\dot{U}_{1bd}$ | $\dot{U}_{1cd}$ |
|---|---|---|---|---|
| Phase angle | 253.4 | −0.2 | 239.8 | 119.8 |
| The judgment of faulty phase | | a, b | | |

The $\Delta \dot{I}_1$ phase angle is between the phase angles of the phase a and phase b relative voltage to the ground, so it can be judged that phases a and b are faulty. The calculation results by substituting the sampled value into Equation (18) are shown in Table 8.

**Table 8.** Calculated insulation parameters of branch 1.

| Insulation Parameters | $C$ | $G$ | $G_{ad}$ | $G_{bd}$ |
|---|---|---|---|---|
| Numerical value | 0.150 | 0.0088 | 0.54 | 2.105 |
| Calculation error | −15% | 17.3% | 8% | 5.25% |

The calculation results show that the insulation of the line is seriously degraded. At this time, the alarm signal of the monitoring system should be issued, and the cable should be replaced. The simulation results of the other branches when they are selected as the faulty branch in turn are shown in Table 9.

**Table 9.** Simulation results for branches 2–4.

| Faulty Branch | Branch Number | Calculated Results of the Pattern Recognition | | Phasor | Phase Angle | $G_d$ (µs) | Calculation Error of $G_d$ |
|---|---|---|---|---|---|---|---|
| | | $C_i$ (µF) | $G_i$ (µS) | | | | |
| 2 | 1 | 0.181 | 0.0074 | $I$ | 0 | 0.5 | 0 |
| | 2 | −0.211 | −0.123 | $U_{ad}$ | 0 | | |
| | 3 | 1.301 | 0.054 | $U_{bd}$ | 240 | | |
| | 4 | 2.314 | 0.095 | $U_{cd}$ | 120 | | |
| 3 | 1 | 0.181 | 0.0075 | $I$ | 0 | 0.5 | 0 |
| | 2 | 0.362 | 0.014 | $U_{ad}$ | 0 | | |
| | 3 | −0.728 | −0.101 | $U_{bd}$ | 240 | | |
| | 4 | 2.314 | 0.097 | $U_{cd}$ | 120 | | |
| 4 | 1 | 0.181 | 0.0076 | $I$ | 0 | 0.499 | −0.2% |
| | 2 | 0.362 | 0.015 | $U_{ad}$ | 0 | | |
| | 3 | 1.301 | 0.055 | $U_{bd}$ | 240 | | |
| | 4 | 1.741 | −0.122 | $U_{cd}$ | 120 | | |

The above simulation shows that the insulation diagnosis for each branch corresponds to the simulation settings, and the insulation parameter calculation error is small. All simulation results proved that this method can quickly identify the faulty cable and fault phase without a lot of calculations.

## 4. Conclusions

A calculation method of cable insulation state based on a decision tree is proposed in this paper. This method judges the line with asymmetrical insulation degradation according to the negative calculated insulation parameters, and it identifies the phase with asymmetrical insulation according to phase angle relationships. Different insulation types correspond to different calculation formulas for the insulation parameters, so the accuracy of the calculated results could be guaranteed. The simulation results show the advantages of this method: it can correctly recognize the insulation type and realize the fault line and phase selection with small calculation errors for the line insulation parameters. However, this paper does not consider the error that may occur when synchronously acquiring zero-sequence current and voltage, as well as the error that a signal conditioning circuit may cause. How to improve the method for gathering and processing signals to reduce errors will be the emphasis of further research to optimize this method. In addition, with the development of computer technology and communication technology, it becomes feasible to apply machine learning algorithms to underground coal mine cable monitoring. Our future research direction will focus on this point.

**Author Contributions:** C.F. proposed the research topic, designed the model, performed the simulations, and analyzed the data. P.Y. was responsible for guidance, proposing the research topic, giving constructive suggestions, and revising the paper. Y.S. performed the simulations and analyzed the data. J.L. improved the manuscript and corrected the spelling and any grammar mistakes. X.Z. and C.S. collected information and organized and revised the manuscript. All authors have read and agreed to the published version of the manuscript.

**Funding:** This research received no external funding.

**Institutional Review Board Statement:** Not applicable.

**Informed Consent Statement:** Not applicable.

**Data Availability Statement:** The original contributions presented in the study are included in the article. Further inquiries can be directed to the corresponding author.

**Conflicts of Interest:** The authors have no conflict of interest to declare. The authors do not have any personal circumstances or interest that may be perceived as inappropriately influencing the representation or interpretation of the reported research results.

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
