# Peer review of "Decision-Making Method for Mine Cable Insulation Monitoring and Grounding Fault Diagnosis"

_processes, doi:10.3390/pr11030795_

Round 1

Reviewer 1 Report

The authors propose a novel classification method based on the decision tree algorithm to reduce the complexity of cable insulation state classification.

Questions:

1. The article needs a better bibliographic revision. The vast majority of articles are over 10 years old. I suggest evaluating more recent articles from the last 5 years. In addition, it is important to point out the weaknesses of existing methods and the benefits that the proposed method aims to solve.

2. Figure 1 is confusing because Figures 1(c) and 1(e) are the same as are Figures 1(d) and 1(f). These figures need better described.

3. Page 2, Line 78: "In the case of single-phase and two-phase deterioration of cable insulation, the equivalent circuit model of the degraded insulation line is different because of the difference of the zero-sequence voltage value in the zero-sequence network." This statement is not consistent with Figure 1 because Figures 1(c) and 1(e) are the same as Figures 1(d) and 1(f).

4. How do the authors identify the type of fault only with zero sequence values? Because that is not enough. Furthermore, what are the types of absences evaluated by the authors and by the proposed method?

5. Is the proposed method capable of identifying the type and location of the fault when it occurs in any position of the cable?

6. Figure 3 is not very clear.

7. Why did the authors use decision trees for this classification problem and not other machine learning techniques?

8. The authors do not present comparative results with methods that already exist in the literature. It would be interesting to present comparative results to evaluate the benefits and weaknesses of the proposed method. Authors can evaluate the results with other machine learning techniques such as Neural Networks or Support Vector Machine.

9. The authors use decision trees but do not report how much data was used for the training and testing steps. In addition, they do not use metrics such as accuracy to assess the results achieved.

Author Response

review report 1

Questions:

  1. The article needs a better bibliographic revision. The vast majority of articles are over 10 years old. I suggest evaluating more recent articles from the last 5 years. In addition, it is important to point out the weaknesses of existing methods and the benefits that the proposed method aims to solve.

Authors’ Response: We would like to thank the reviewer for the suggestion.

The references have been updated and the literature review and status analysis have been revised in response to comments.

  1. Figure 1 is confusing because Figures 1(c) and 1(e) are the same as are Figures 1(d) and 1(f). These figures need better described.

Authors’ Response: We would like to thank the reviewer for the suggestion.

We are very sorry that we inserted the wrong picture during typesetting due to our mistake. The picture error has been corrected.

  1. Page 2, Line 78: "In the case of single-phase and two-phase deterioration of cable insulation, the equivalent circuit model of the degraded insulation line is different because of the difference of the zero-sequence voltage value in the zero-sequence network." This statement is not consistent with Figure 1 because Figures 1(c) and 1(e) are the same as Figures 1(d) and 1(f).

Authors’ Response: We would like to thank the reviewer for the suggestion.

We are very sorry for this problem caused by the error of our picture. We have made changes.

  1. How do the authors identify the type of fault only with zero sequence values? Because that is not enough. Furthermore, what are the types of absences evaluated by the authors and by the proposed method?

Authors’ Response: We would like to thank the reviewer for the suggestion.

In fact, due to the extremely harsh safety and working conditions in the coal mine, the monitoring means that can be used are extremely limited, and the data that can be obtained depends on the low current grounding system. Therefore, the only data that can be used for the calculation is the zero-sequence current voltage at the end of each cable. The goal of this paper is to accurately judge the coal mine cable grounding fault caused by insulation decline under the condition of only data, select and alarm the fault cable, and prevent the coal mine safety accidents caused by cable ground fault. The goal of this paper can be achieved through these data. Therefore, this paper only calculates the cable fault through the listed zero-sequence network. Other fault types of coal mine cables, such as partial discharge, are not the focus of this paper. For this suggestion, an explanation has been added in the article, and part of the description has been modified to make the goal of the article clearer.

  1. Is the proposed method capable of identifying the type and location of the fault when it occurs in any position of the cable?

Authors’ Response: We would like to thank the reviewer for the suggestion.

As mentioned above, the most urgent problem of power supply in coal mine is to judge the faulty cable and cut it off under the condition of extremely limited monitoring means. This method can select the faulty cable or even the faulty phase, but it cannot identify the location of the cable fault. For cable fault location, this is a very good problem and is what we are working on in the next step.

  1. Figure 3 is not very clear.

Authors’ Response: We would like to thank the reviewer for the suggestion.

We have reformatted Figure 3 to make it clearer.

  1. Why did the authors use decision trees for this classification problem and not other machine learning techniques?

Authors’ Response: We would like to thank the reviewer for the suggestion.

This is a very good question. At present, the most urgent problem of power supply in coal mine is to quickly judge the faulty cable and cut it off under the condition of extremely limited monitoring means. However, due to the extremely harsh underground safety and working conditions, the monitoring means that can be used are extremely limited. The data that can be obtained depends on the low current grounding system, and the amount of monitoring data that can be obtained is also limited. Moreover, the current underground cable monitoring system is difficult to use other machine learning algorithms which require large-scale calculation because the conditions now generally use DSP as the calculation core. The classification criteria used in the decision tree algorithm used in this paper are not iterated by the machine learning algorithm mining, but derived from the expert experience and through the rigorous formula derivation and analysis calculation. This is a fast and effective way of making judgements. Of course if more accurate and effective machine learning algorithms can be used, this is a very good option and is one of the directions we are currently focusing on.

  1. The authors do not present comparative results with methods that already exist in the literature. It would be interesting to present comparative results to evaluate the benefits and weaknesses of the proposed method. Authors can evaluate the results with other machine learning techniques such as Neural Networks or Support Vector Machine.

Authors’ Response: We would like to thank the reviewer for the suggestion.

Some changes have been made to the simulation verification section of the article. The verification session of this paper is to prove that the proposed method is feasible and effective by means of simulation verification. Moreover, our ongoing work is to apply the proposed method to the field environment of coal mine to further verify the feasibility and accuracy of the proposed method. Due to the special operating environment of coal mine cables and the small number of fault samples that can be obtained at present, it is difficult to use neural network or support vector machine method for comparison and verification. However, by breaking through the current technical bottleneck, applying machine learning algorithms to the fault diagnosis of coal mine cables is a focus of our next research direction.

  1. The authors use decision trees but do not report how much data was used for the training and testing steps. In addition, they do not use metrics such as accuracy to assess the results achieved.

Authors’ Response: We would like to thank the reviewer for the suggestion.

Due to the particularity of coal mines cables, it is difficult to obtain large data samples for the training of machine learning algorithms. The classification judgment indexes in the decision tree method we used is all calculated by expert knowledge analysis. All parameters are strictly deduced according to the formula. Therefore, the index parameters should be precisely fixed values, which can be referred to the formula deduction process in Section 2 of this paper. Therefore, the simulation example is to use the simulation verification experiment to verify the feasibility and accuracy of the model.

We would like to thank this reviewer again for the valid and constructive comments which helped us improve the quality of our paper. We hope the revised manuscript meets your expectation.

Reviewer 2 Report

The paper presents the study about decision making method for mine cable insulation monitoring and fault diagnosis. Authors say, that  online monitoring of the power cable state has significance for ensuring the safe and economic operation of distribution systems. They developed a classification method based on the decision tree algorithm. Authors used zero-sequence network, and the positive and negative values were generated to identify if the cable insulation was symmetrical. The value of presented method was verified using mine power distribution system model. Obtained results proved that the method accurately diagnoses the power cable insulation state.

Dear author, thank you very much for interesting paper about method of high voltage cable diagnostics. It is still important problem, especially in case of mine cables. I put some comments and questions.

Comments:

1. The introduction chapter is well organized. Anyway, the paper describes a diagnostic method of high voltage cables, so I would expect more information about other diagnostic methods in order to compare positive and negative aspects of proposed method.

2. Authors present method which should detect some problem in cable insulation. Please present more information about eventual problems which may occur in high voltage cable insulation.

3. Presented method compare total reactance, conductance of cable insulation. Please explain why it is correct approach. Some defect (partial discharges) occurs in small distance of cable, and it can be dangerous, but total reactance and total conductance may differ a little comparing to good cable. How this problem is solved by authors.

4. there are some information about signal processing. Please give us some information about the signal. It is generated signal, or collected signal as results of partial discharges for example.

Author Response

review report 2

Comments:

  1. The introduction chapter is well organized. Anyway, the paper describes a diagnostic method of high voltage cables, so I would expect more information about other diagnostic methods in order to compare positive and negative aspects of proposed method.

Authors’ Response: We would like to thank the reviewer for the suggestion.

According to your comments, we have revised the introduction section extensively, adding the description of other diagnostic methods and elaborating their advantages and disadvantages.

  1. Authors present method which should detect some problem in cable insulation. Please present more information about eventual problems which may occur in high voltage cable insulation.

Authors’ Response: We would like to thank the reviewer for the suggestion.

According to your comments, we have revised the article to clarify the problems and objectives to be solved by this research.

  1. Presented method compare total reactance, conductance of cable insulation. Please explain why it is correct approach. Some defect (partial discharges) occurs in small distance of cable, and it can be dangerous, but total reactance and total conductance may differ a little comparing to good cable. How this problem is solved by authors.

Authors’ Response: We would like to thank the reviewer for the suggestion.

At present, the most urgent problem of power supply in coal mine is to quickly judge the faulty cable and cut it off under the condition of extremely limited monitoring means. The aim of our research is to accurately judge the coal mine cable grounding fault caused by insulation decline, select and alarm the fault cable, and prevent the coal mine safety accident caused by cable grounding fault under the condition of only data. Under such conditions, it is difficult to detect partial discharge. Our proposed method can only detect cable insulation deterioration when it is relatively severe. This method is based on expert experience and knowledge through rigorous derivation and analytical calculation, in the Section 2 of the article in detail the derivation process, and finally through the simulation verification experiment to verify the accuracy of this method.

  1. there are some information about signal processing. Please give us some information about the signal. It is generated signal, or collected signal as results of partial discharges for example.

Authors’ Response: We would like to thank the reviewer for the suggestion.

The signal data used for verification in the paper was generated by MATLAB software and is a common current signal. In the practical engineering application, the underground cable monitoring system in coal mine only has zero-sequence current grounding monitoring system, and the signals that can be collected are only zero-sequence current, zero-sequence voltage, three-phase voltage and other power frequency signals. However, it is difficult to collect the tiny partial discharge signal because of the complex operation environment of the coal mine cable. This is a very good question and we will focus on it in the next step.

We would like to thank this reviewer again for the valid and constructive comments which helped us improve the quality of our paper. We hope the revised manuscript meets your expectation.

Round 2

Reviewer 1 Report

The authors propose a novel classification method based on the decision tree algorithm to reduce the complexity of cable insulation state classification.

The article has been improved, the contribution is good and all questions have been effectively answered.